# The Impact of Non-Physical Education Teachers’ Perceptions on the Promotion of Active and Healthy Lifestyles: A Cross-Sectional Qualitative Study

**DOI:** 10.3390/ijerph20032026

**Published:** 2023-01-22

**Authors:** Lúcia Gomes, João Martins, Madalena Ramos, Francisco Carreiro da Costa

**Affiliations:** 1CIDEFES, Universidade Lusófona, 1749-024 Lisbon, Portugal; 2Centro de Estudos em Educação, Faculdade de Motricidade Humana e UIDEF, Instituto de Educação, Universidade de Lisboa, 1649-004 Lisbon, Portugal; 3CIES Iscte, Iscte-University Institute of Lisbon, 1649-026 Lisbon, Portugal

**Keywords:** active lifestyles, teachers’ past experiences, physical education

## Abstract

(1) Background: In accordance with the socio-ecological model of physical education (PE), school-based interventions to promote physical activity (PA) will only be successful if a change occurs in the perceptions and attitudes of all. This study sought to analyze non-PE teachers experiences in PE and how these experiences relate to the value they attribute to PE, sports and PA, and the impact of teachers’ perceptions on promoting active lifestyles in the school’s context. A mixed-method study was adopted. (2) Methods: A survey about the perceptions in PE was applied to 297 teachers (58 male). From this sample, 24 teachers were selected for three focus groups considering their experiences in PE (e.g., good experiences, bad experiences). MaxQda was used for the inductive qualitative data analysis. (3) Results: Teachers who have had good experiences in PE value the role of PE and sports. Teachers who have had bad experiences do not value the importance of PE or sports. (4) Conclusions: The results support the importance of having good experiences in PE to generate positive attitudes. Thus, teachers who have had bad experiences in PE may constitute barriers that will hinder the promotion of an educational climate that promotes PA and active lifestyles in schools.

## 1. Introduction

There are well-documented health benefits associated with regular physical activity (PA) [1,2,3]. School and physical education (PE) have been considered as ideal contexts regarding the promotion of active and healthy lifestyles, both by researchers [4,5,6] and by numerous organizations [7,8], taking into account that the majority of children and young people from all social, cultural and economic groups attend school and spend a considerable amount of time there. A whole-school approach [9] is necessary to promote an atmosphere in which PE, PA and sport are valued and encouraged by all members, recognizing that all aspects of the school community can impact on students’ health and well-being, and bearing in mind that more effective schools are characterized by a shared vision of teachers and principals, acting on behalf of the learning and education of all students [10].

In accordance with the socio-ecological model of school [11] and PE [12], school-based interventions to promote young people’s PA will only be successful if a change occurs in the perceptions and attitudes of politicians, school principals, teachers, and parents toward the value of PE, sport, and PA. One of the strongest lines of research on school change is the work related to teachers’ reform efforts. Research suggests that a teacher’s values, beliefs, and perceptions will influence the way reforms are interpreted and implemented [13,14,15]. Teachers decide and act according to their own sense of identity, educational orientation, and the meanings they link with particular ideas and concepts around them. These meanings and ways of understanding oneself and the world constitute an individual’s subjectivity. Through a range of discursive practices and experiences, teachers develop stable emotional characteristics, beliefs, and values that contribute to the structure of their activities and professional decisions [16,17,18]. Teachers from all curricular areas have a range of subjective attitudes towards PE and PA, built when they were primary and secondary school students [19]. These prior experiences that are remembered in a positive or a negative way can exert influence in their beliefs about the role and purpose of the PE subject. Therefore, teachers, especially primary teachers and secondary classroom teachers, can function as barriers or facilitators in promoting active lifestyles among the school population [20].

In Portugal was carried out a study on teachers of all subjects, students, and parents in six of Greater Lisbon’s basic and secondary schools. Two hundred and ninety-four teachers participated in the study (75 males, 219 females, 40 PE teachers, and 254 teachers of other subjects) [21]. One of the study’s objectives was to characterize teachers’ views regarding PE taking into account their past experiences with PE. The teachers were divided into three distinctive groups. Teachers with a positive experience, evaluating their past experience as good or very good; teachers with a neutral experience, evaluating their past experience as neither good nor bad; and teachers with a negative experience, evaluating their past experience with PE as bad. Only variables related to the teaching–learning process distinguished the groups of teachers with positive and negative past experiences in PE. The teachers who evaluated past experiences in PE as positive rated the characteristics of the PE in terms of relationships with peers, learning benefits, contents and the competence of the PE teachers as “very important”. The group of teachers who had a bad or very bad experience with PE referred to the characteristics of the PE classes and contents as variables that were extremely important in forming their perception. It was also verified that most classroom teachers need an adequate idea of the real needs of children’s PA engagement in accordance with the WHO recommendations regarding PA. Furthermore, a dominant socio-cultural view of PE as peripheral to the accomplishment of the central functions of schooling was identified among most of these teachers. Therefore, teachers’ perceptions are probably influenced by their past experiences in PE, given that a significant proportion of teachers (37.8%) responded that they had not had a positive experience in PE as a student. Unfortunately, within many educational communities, the demand for more time for subjects believed to constitute the fundamental curriculum takes time away from subjects considered non-vocational (e.g., PE, music and arts).

Obtaining the support and adherence of all educational agents is crucial to success in school interventions aiming to promote the practice of PA; however, little is known about the role of secondary classroom teachers. (In Portugal, during the first cycle of education (6–9 years), PE is the responsibility of the classroom teacher, a generalist with a master’s degree in basic education. He or she has the autonomy to decide when and how much time dedicated to PE per week. In the second cycle of education (10–11 years), third cycle (12–14 years), and secondary level (15–17 years), PE is imparted by a teacher with a master’s degree in PE and School Sport.) In promoting PA and sports, this study sought to explore and analyze teachers’ past experiences in PE and how these experiences relate to the value they attribute to PE and PA. To understand the status that PE has in the school setting, it is important to identify what all teachers, mainly classroom teachers, think about the PA process and the promotion of PA at schools. In addition, the present study intends to shed light on attempts to understand the impact of teachers’ perceptions on promoting active lifestyles in the school context.

## 2. Materials and Methods

### 2.1. Study Design and Study Population

To find in-depth descriptions of the perceptions, attitudes, and behaviours of different teachers, a mixed method was used. First, a validated survey [22] was applied to a convenient sample between January and February of 2015; 297 teachers (58 males and 239 females) from two Portuguese secondary public schools participated. The survey was about experience in PE, knowledge about PA guidelines, and perceptions and beliefs of the aims and assessment in PE. For the intensive part of the study–which is our focus of this paper–24 teachers (5 males and 19 females, with a mean age = 55.7, max age 62; min age 34; 86 math and sciences teachers, 86 languages teachers, 51 arts teachers and, 50 history, philosophy, psychology and geography teachers) were selected considering two criteria from their answers (i.e., experience in PE and PE status). For focus groups, teachers were organized considering their early experiences in PE (i.e., good experience; bad experiences; or neither good nor bad experiences) as well as their perceptions about the status of PE in school curriculum (mandatory with evaluation; mandatory without evaluation; optional; or no opinion). Group one comprised six women and one man, between 44 and 56 years old, two Portuguese teachers, three math teachers, one philosophy teacher, one history teacher and one science teacher, with good past experiences in PE and favourable perceptions of PE (FG “+”). Group two incorporated seven women and one man, between 35 and 62 years old, two from native language, three from math, two from philosophy and history and one from sciences, with negative past experiences in PE and a unfavourable perceptions about PE (FG “−”). Group three was a heterogeneous group. The third group consisted of teachers with positive experiences and favourable perceptions as well as teachers with negative experiences and unfavourable perceptions towards PE (FG “±”), six women and two men, between 34 and 59 years old, one Portuguese teacher, four math teachers, one geography teacher, one history teacher, and one science teacher. In addition, ta study was carried out with a pilot group with the same characteristics.

### 2.2. Procedures

The focus group was part of a larger study (a mixed-method study). Therefore, a prior script was made and discussed among the research team, considering the aims of the study. Then, the script was submitted to a group of research experts in PE and Portuguese language for validation or to suggest changes. Next, a pilot study with a sample with similar characteristics to the population (8 teachers, 5 with positive perceptions, 3 with negative perceptions; mean age 54.3). The goal was to train the researcher in conducting the focus group, test the script and improve it.

The questions to the focus group were mainly drawn from the questionnaire data (building the script together with other literature is not the object of this paper, but served as a basis) and the gaps in knowledge identified in the literature review, considering the study’s objective and exploratory design. The final focus group script was composed of eight questions: two preliminary questions about what school they had attended in childhood and youth, and another about their past experiences in PE. A second group of five questions aimed at a deeper understanding of how these experiences influenced their current lifestyle, perceptions about the status of PE, and ranking of PE in school curricula. Further questions were about PE assessment (mandatory or not), PE aims, and, ending this set of questions, one question about the role of the school in promoting and developing active lifestyles. Focus group questions are available in Table 1.

All the participants gave their consent in writing prior to their participation, and the purposes of the research were laid out to them. They were told the focus group would be tape-recorded and that any information they gave would be kept confidential, that they were also free to leave each section at any time. Personal data from this study was anonymized using identification by numbers in the survey and nicknames in the focus groups. Each of the three focus groups took place in a quiet room in the school, for approximately 50 min, in June of 2015. One researcher monitored the discussions in each of the three focus groups (and trial) and took notes during interactions. A small snack was served to each focus group at the end of the session. All focus groups were audio-taped and transcribed verbatim. Ethical approval was granted by the University committee. Schools and teachers agreed to participate in the study.

### 2.3. Data Analyses

Focus groups were transcribed verbatim for data analysis using Microsoft Word. During the transcription, some units of recording (phrases/words), were highlighted in colour, which seemed relevant to the study purposes. After transcribing the focus groups, the protocols were again examined to confirm and check for any missing information. This was an opportunity to highlight other units or recordings through the audio records of the focus group, audio files, written transcripts, and field notes. This was an iterative process that occured simultaneously [23]. A thematic analysis [24] was carried out, in which the data was analyzed inductively. First, transcripts were read and reread. Using an iterative process, meaningful quotes were then identified. A continuous comparison method was implemented, in which quotes were continuously compared with one another. Two more investigators helped when there was some doubt. Citations considered to represent the same meaning were grouped together and allocated a label, while quotes considered to represent a different perception were given a new label. All data were examined until all significant data had been identified, clustered, and labelled. The resulting labels were then scrutinized and organized into themes, each theme comprising labels considered conceptually similar.

Subsequently, the documents with the focus group transcripts were imported into the MAXQDA 11 (VERBI, Berlin, Germany) content analysis program. During the investigation process, this instrument was updated to MAXQDA 12 (VERBI, Berlin, Germany), allowing the treatment of data for focus groups; this facility did not exist previously, and helped in the final process analysis.

Initially, only two focus groups were analysed, based on the thematic analysis procedures [24] comparing common themes in the two focus groups. This depended on the ideas expressed in each question, which could be words, phrases or parts of sentences, and paragraphs. In each recording unit or context, mutual exclusion of categories took place, taking into account accuracy and consistency [25] to encode, create categories, filter and question the information. This was in order to integrate the information and give it a logical sense in responding to the research questions.

Throughout the process, the categories were successively compared and re-examined throughout the process, considering the aim in question. Subsequently, the categoric system that was developed was subjected to validation by two research experts. As the result of the analysis of each question, the following category system was created, composed of three dimensions of analysis: (i) dimension of psychological factors and cognitive factors; (ii) social factors; (iii) educational factors related to PE; each one with categories and subcategories (Table 2).

## 3. Results

### 3.1. Previous Experiences in Physical Education

Through experience, teachers develop stable emotional characteristics, which contribute to structuring their beliefs (e.g., aims, status) [16,17,18], attitudes, (i.e., knowledge about the guidelines of PA for themselves and for students; knowledge about the influence of PE and Sport on academic achievement), and behaviours, (e.g., classes; teaching council), helping them in their pedagogical activity.

In this study, teachers who had good experiences with PE in elementary and secondary school value the role of PE in the promotion of PA. According to them, the factors that contributed to build good experiences were the aspects related to the planning and organization of the class, which are the responsibility of the PE teacher (e.g., technical competence and student-teacher relationship):


*And that was where I liked it even more, because I had a very active PE teacher.*
“FG “+” (Fem1)


*She (PE teacher) was very good, explained everything.*
“FG “+” (Fem1)

(i)Dimension of psychological and cognitive factors (good or very good experiences)

The first dimension identifies individual factors related to perceptions, attitudes, and knowledge (experiences, perception of competence and attitudes).

Perception of competence–refers to feelings of ability to perform some physical tasks.


*Well, for me it was easy, I could do everything.*
“FG “+” (Male2)

Attitudes towards PE–favorable references related to PE (e.g., like, fun).


*(...) right, I really liked it, always, it was a subject that I always put effort on, because I liked.*
“FG “+” (Male1)

Meaningful PE can help students value their experiences in PE and understand how participation improves the quality of their lives. Moreover, it emphasizes the importance of the development of meaningful experiences in the early years of schooling for children and young people, contributing to more positive attitudes towards PE, sports, and PA, given it is a favorable period for structuring active live styles [26,27,28]. Teachers should prioritize strategies that support students’ autonomy, reflection, and goal-setting [29].

(ii)Dimension of social factors (good or very good experiences)

The second dimension is related to the influence of family and friends on PA participation.

*Family*–favorable references related to family members.


*I started going to soccer with my father, ah... I began to like it.*
“FG “+” (Fem3)

*Friends*–favorable references related to peers.


*I used to ride bike with my friends, and I still do it.*
“FG “+” (Male2)

Grounded in social learning theory [30], students’ behavior and behavioral sets are likewise affected by social models. References in the literature [31,32] indicate that there is a consistent and positive association between the providing of social support by family (e.g., father, mother, and siblings) and friends, and more engagement in PA [31,33]. To this point, schools can play a significant role in strengthening the importance of PA and sports, not only with students, but also with their families and community (PE teachers in partnership with other subject teachers can also take this approach, if their experiences are associated with good memories in PE) [9].

(iii)Dimension of educational factors of PE (good or very good experiences)

The third dimension refers to the identification of past experiences in PE classes, relating to the curriculum, the teacher, the class characteristics, and the resources.

Curriculum–favorable references related to subject matter in classes (i.e., sports games and competition for male teachers, and gymnastics and dance for women).


*But I remember that I loved to dance.*
“FG “+” (Fem2)


*But what I really liked was the sports, soccer... And the competition … I love the competition.*
“FG “+” (Male1)

Teacher–favorable references related to ability and personality of the teacher (i.e., kindness and patience).


*He had a way of motivating us and even if we did not like it, he would adapt and we would do it willingly.*
“FG “+” (Fem4)

This highlights the role of meaningful interactions between the teacher and the curriculum learning context, identifying how teachers’ actions and class participation can lead to better experiences. Moreover, it recognizes that the encouragement received from the teacher enhanced meaningful engagements with PE and PA [34].

For this reason, we underline the importance of clearly identifying the objectives of each task, their importance, and usefulness. Students must understand the meaning and real value of what they are learning, providing them with clear opportunities to appropriate this knowledge, reflect on their experiences and the ability to accomplish them [35].

There is a belief that men prefer playing team invasion games, mainly the typical European sport (i.e., soccer) with friends or peers, and that includes groups of the same ability level, considering the need they have to compete [26]. For male teachers, competition is mostly associated with good experiences. On the other hand, for teachers with negative experiences, the concept of winning or losing was not recalled. Considering this, PE teachers must be careful about how they plan; providing situations that take the focus away from the competition and performance improvement [36]. Establishing a climate of mastery using the target model that emphasizes activities involving perceptions of task-involving relatedness, pleasure, cooperation, the need for satisfaction, and self-determined motivation (i.e., intrinsic motivation), could be a good strategy [37].

For female teachers, their perceptions of PE and PA are specially related to fitness and health rather than games or competition. Female teachers’ preferences fall into individual activities (i.e., dance, gymnastics). Probably this is because they have more opportunities for participation and more self-awareness of competence. Several authors [38,39] argue that sports, especially games, have been culturally defined, structured, encouraged, and built as a male occupation. Being a male-dominated culture makes them feel less feminine, and it could turn females away from participating [40,41,42]. Some female teachers have reported embarrassment and low perceived ability which induced a lack of interest in competitive activities. The dominance of males in sports is seen as an obstacle to women’s involvement.

Dedicated literature suggests several approaches to improve girls’ participation in PE: considering women’s participation in future PA, reflecting girls’ voices and needs [43], ensuring inclusive participation, modifying scoring in mixed-gender activities, considering group strategies and preferences to encourage their participation, developing their capacity for communication and negotiation, enhancing insightful and continuous involvement, promoting opportunities for transformation and sense of ownership in learning, promoting better perceptions about PE and sports [44,45,46], single-gender classes [28], the use of digital tools (i.e., Instagram) as another helpful learning space for empowerment for girls [47], and the use of role models [48].

Regarding teachers who have had bad experiences in PE, they do not value the importance of PE and PA in the global development and education of students. The factors they list for negative attitudes toward PE include: traumatic experiences, a low level of PE teacher competence, curriculum, and the use of specific equipment.

(i)Dimension of psychological and cognitive factors (bad or very bad experiences)

Traumatic experiences


*(...) For example, when I did some somersaults in the plinth, for me, it was a fright; I was terrified, scared, and I only did it because they forced me.*
“FG “−” (Fem4)

Perception of competence


*Because I was awful in PE, I could do nothing, and the teacher sent me to be the referee.*
“FG “−” (Fem2)


*I don’t know how to swim and my teacher pushed me into the water.*
“FG “−” (Fem4)

Adults have extraordinarily vivid negative memories of PE from their youth; reports were associated with poor perceptions of competence and relationships, two basic psychological needs assumed in the self-determination theory [49], which are possible determinants of subsequently diminished PA behavior.

Moreover, support activities where pleasure, mastery, well-being, and fun were established to achieve PE aims are considered essential indicators of lifelong participation in the culture of movement and are strictly linked to learning [50].

(ii)Dimension of educational factors of PE (bad or very bad experiences)

Curriculum


*The classes always started the same … running.*
“FG “−” (Fem3)

PE teacher


*Some teachers are bad leaders for students.*
“FG “−” (Fem1)


*He made us run around the court, while he sat on a bench, reading a newspaper.*
“FG “−” (Fem2)

Specific equipment *(as well as its transportation)*


*I did not like PE days because I had to walk loaded with... with.... with track suits, right? ... and we had to walk all day with the bag, it was more weight.*
“FG “−” (Male3)


*I searched my father’s drawer to find the biggest sweater to wear.*
“FG “−” (Fem1)

These personal experiences are typically formed with social support from teachers or colleagues, which can enhance meaningful engagement with content [26].

Teachers mentioned curriculum blocks of sports in PE classes and teacher-directed learning as the concepts used by their PE teachers they remember unfavorably. The references to ‘running’, associated with ‘being’ healthy or having better physical condition, neglects other ways of viewing PE in a more holistic way; some authors call this a narrow way of seeing PE [28].

Also, some teachers with bad experiences mentioned that changing clothes in the locker room was uncomfortable. Others pointed out that they felt on display in classes and constantly judged by their body types. These incidents created negative past-performance experiences.

Moreover, teachers who formed unfavorable reports about their experiences also highlighted the aspects related to the benefits collected:


*It was good for my health and it was also fun.*
“FG “−” (Fem3)

Having fun must be considered a significant part of their experience in PE or sports, but it does not mean undisciplined or disorganized classes. Fun is generally associated with absence from learning, and students can regard PE as not serious [51]. Although, as some authors [26] mentions, fun should not be considered the only criterion for meaningful experiences, but a way to develop them.

### 3.2. Status of PE

In 1978, UNESCO granted the “fundamental right” status to PE and sports.

In addition, it established that each education system should guarantee its practice for every student, being linked to other educational components. Moreover, it considered PE the most proficient way to provide all children and youth with the skills, attitudes, knowledge, values, and full competency for necessary lifelong involvement in society. However, for decades, PE has been seen as a subject that only emphasizes physical or practical issues, with a limited focus on cognitive or educational content [52]. Although PE has become a feature of national curricula in many countries, it still struggles for legitimacy and is not taken seriously by all members of staff [53]. It is known that school experiences in PE shape teachers’ commitment and interest in PA or sports [36]. Therefore, these experiences can help build interactions and bonds, providing a support network and promoting the values of PE and an active lifestyle.

Teacher’s perceptions about PE aims.

Given the meaning of lifelong learning and holistic educational outcomes (i.e., the required learning goals that schools and teachers and other school staff hope students accomplish), according to the Portuguese student profile at the end of secondary school [54], PE is one of the key learning areas that could educate students through PA. Literature links PE aims and benefits with personal development, social skills, and academic achievement, with health promotion a central focus worldwide [55].

Promoting a quality PE (QPE) may encourage students to embrace a culture of movement, competitive sports, or other physically demanding activities. High quality physical education is associated with the development of physical and psycho-social well-being, peer-led learning and supporting students in developing physical, social, and emotional skills that define healthy, resilient, and socially responsible citizens. QPE grants equal opportunities for all students to become physically confident and encourages an active and healthy lifestyle; the aims and goals of PE are a guarantee [56].

Identifying the perceptions of other subject teachers about the aims and goals of PE is determinant to knowing their beliefs about it [18].

Teachers who had good PE experiences considered PE a way to promote good ethics as well as important for promoting healthy lifestyles.


*It’s important to promote good ethics and healthy lifestyles. Helps in development of good citizens.*
“FG “+” (Fem2)

Teachers who had very bad PE experiences considered PE as a means of catharsis, as well as important for promoting healthy lifestyles.


*PE should be something to promote catharsis when you are tired.*
“FG “−” (Fem5)

It would be important that all PE teachers consider explaining the goals and values of the discipline, not only because it has an educational purpose, but also because it values the profession. On the other hand, it may help to demystify the preconception associated with the standard of academic success if it focuses exclusively on academic subjects (i.e., mathematics and native language), enhancing the favourable aspects expressed in the research of the benefits of PA on students’ academic achievement. The support of all educational agents promoting the practice of PA is essential to success in school interventions [57].

Teacher perceptions about assessment in PE.

Teachers who had very bad PE experiences considered that PE should be mandatory without assessment:


*I think it should not be considered … (assessment), however should be mandatory.*
“FG “−” (Fem4)

On the other hand, teachers who had very good experiences in PE, thought that PE should be mandatory with assessment:


*If it is a school subject, it must be mandatory and must be assessed like others…to be taken seriously.*
“FG “+” (Fem4)

Assessment is a pedagogical tool in the learning process. It should involve goal setting and explanation, as well as formative tasks aiming to: (i) inform students and teachers about where they are in their learning process; (ii) establish where they intend to go; and (iii) help them to adjust and understand what can be done to achieve the final goal. However, the assessment of learning usually only plays a summative purpose in which students show their knowledge and are given a grade/result associated with their level of success. Moreover, the lack of explanation, goal setting, and appropriate assessment tools in PE probably makes a teacher look more focused on final skills, knowledge acquisition, and grades Literature suggests a roadmap for learning, which means offering different levels of achievement [58]. At each level on the roadmap, the assessment should support students with feedback about their progress, focus on what they have already accomplished, and establish new strategies to step up to another level of skill and knowledge. Portugal’s PE scholar system already provides a program articulated horizontally by year and vertically by years (i.e., skills, knowledge, and fitness achievement). However, PE teachers still maintain their idiosyncratic modus operandi. Furthermore, instead of applying new knowledge, they maintain old practices acquired in their time as a student [59].

Other subject teacher perceptions about school responsibility in promoting active and healthy lifestyles.

Emerging studies [60,61] indicate that the responsibility for promoting active and healthy lifestyles should be a political decision, involving organizations and their choices (i.e., schools) as well as adult role models (i.e., teachers, parents). Some authors [62,63] suggest that the perception of responsibility can be seen as a moral attribute (being responsible) or as an action (acting responsibly). Other subject teachers’ perceptions strengthen this point of view, saying that the school responsibility is:


*Providing opportunities and educating for active lifestyles.*
“FG “+” (Male3)

Alternatively, providing more time for activities in the school or spaces where students can play:


*I really think that schools should create more... more space, and when I say space, I mean more moments (…) but on the other hand, also create more opportunities, recognizing that there are plenty of spaces and activities in school, through sports in school.*
“FG “+” (Male1)

However, other participating non-PE teachers that have negative perceptions of PE and PA only mentioned that the school should provide opportunities for being active. However, they considered that this responsibility lies only with PE teachers:


*I think PE teachers already do this … and they have plenty of activities in annual planification.*
“FG “−” (Male2)

Representing the world through a specialized window is too reductive for each teacher and the school’s roles. The school must be seen beyond its standardized role of learning and teaching in class, and consider all aspects of school life [64]. Promoting active lifestyles should not only be a shared responsibility, but an ethical obligation to provide students with information about the values of PA, in the same way that teachers provide information about other common values (i.e., respect, friendship).

Schools must change; teachers are potentially the most decisive elements in the student’s education, along with families and the rest of the community, with different levels of influence [58]. Investing in teacher training and strengthening their legitimacy by improving their knowledge can facilitate common interventions that promote shared practices to improve quality of life [9].

## 4. Conclusions

The results of this study support the importance of having good experiences in PE to generate positive attitudes towards PE. Thus, teachers who have had bad experiences in PE may constitute barriers that will hinder the promotion of an educational climate that promotes PA and active lifestyles in schools. Unfortunately, a significant number of teachers continue to share the idea that PE and PA negatively influence student achievement. Furthermore, the results also express the need for PE teachers, in addition to their work in the classroom, to interact with other subject teachers, divulging the benefits of PA and, above all, the scientific evidence on the influence of PE and PA on academic achievement and student behaviour.

In line with the socio-ecological model of school [11] and PE [12], we share the idea that any school intervention to promote PA will only be successful if there is a change in the perceptions and attitudes (i.e., politicians, school principals, teachers and parents) towards the value of PE, sport, and PA [9].

## Figures and Tables

**Table 1 ijerph-20-02026-t001:** Focus Group questions.

Preliminary questions	1. Do you remember your primary and secondary schools? Did you have Physical Education (PE)? What is your best memory of it? 2. From your personal experience, how do you evaluate PE in school (or schools) you attended? What of positive and negative experiences do you have from your PE classes? 3. How did these experiences influence your current lifestyle? 4. Some of you considered that PE should (not) be part of the five ranking curricula of students. Can you explain it? 5. On the survey, you stated that PE should (not) be mandatory... would you like to share your views? 6. In your opinion, what should be the aim of PE at school? 7. In your opinion, what responsibility does the school have in promoting active lifestyles? What do you think your school does? 8. Is there anything else would you like to share that has not been asked?
Deepening knowledge	
Final comments	

**Table 2 ijerph-20-02026-t002:** Categories and subcategories of analysis.

Categories	Subcategories
	Psychological and cognitive factors
Previous experiences in PE	Social factors
	Educational factors of PE
Status of PE	Aims
	Mandatory with or without evaluation
School responsibility in promoting	Providing opportunities
active and healthy lifestyles	Educate for active lifestyles

## Data Availability

The authors have data base.

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
