# Peer review of "The Impact of Non-Physical Education Teachers’ Perceptions on the Promotion of Active and Healthy Lifestyles: A Cross-Sectional Qualitative Study"

_ijerph, 2023, doi:10.3390/ijerph20032026_

Round 1

Reviewer 1 Report

Introduction and Theoretical Framework

Although it is true that the subject is of relavancy, there is a lack of literature to justify the study and the need for teachers of other subjects to be involved in these processes.

Material and Methods

The sample of subjects used is not uniform, but is biased, since the mean age is 55.7 years, which is a fact that must be taken into account, since the way of teaching physical education has changed a lot, and therefore the study starts with a bias of origin.

It is strongly recommended that a second, younger group of teachers be sought and that we compare whether their perception as students has influenced their current vision.

Results and discussion

The results are presented in a coherent manner, but the bias in the initial sample means that the results obtained cannot be extrapolated to other fields.

Conclusions 

As has been commented,the sample although diverse in terms of teacher typology, both the age and the questions asked were on the way to seek that those teachers with a bad experience in physical activity could "contaminate" the results and therefore the meetings extracted from them.

Bibliography/References

The references are correct and well used, but it would be necessary to extend them not only to the bad experiences that condition the way of teaching, but also to offer other points of view.

Author Response

Dear reviewers,

We would like to thank you for the constructive comments addressed to this paper.

We have tried to respond positively to all the comments made. Regarding the concrete comments made by the reviewers, please consider the following questions and our answer to them (in blue)

Reviewer 2 Report

I consider the document to be of great interest because it is necessary to insist on research of this qualitative nature, so that educational situations in this area of study can be better understood. However, I also believe that the results should have been "discussed" with more research from the country where the research was carried out. This leads to an understanding of the context and to see what would happen if asked about in other European countries.

Author Response

(The authors gave the same response as above.)

Reviewer 3 Report

This is an interesting article, which explores a number of issues that may be highly relevant to the readership. I have some general comments about the term “past experiences in PE”, the corresponding literature review, and methods/discussion for the author(s) to consider.

Introduction

“past experiences in PE”: I totally agree with you, that prior experiences influence the way how people “see” something. Particularly with regard to PE, PA, and sports, it would be helpful to differentiate between past experiences in PE, past experiences with PA (which could have an impact on PE-experiences as well) in/not in school (school as the environment for PA, sports or a sports club in leisure time). You refer to literature addressing the relation of social support for PA and sports (pg 5, ln 211-216). Within the FG (table 1) you then stress on PE as a subject in school. This should be clearer from the beginning.

This leads directly to the next point: it seems you are kind of reviewing the literature on pg 2, ln 56-76 (?). I think it may relevant to first differentiate what you mean by “past experiences in PE” and after that review the literature. Isn’t there research addressing the role of past experiences in PE? If so, state it, that will strengthen your research.

Pg 2, ln 72: “a dominant socio-cultural view of PE”, can you elaborate a bit more on this as it seems relevant how you see PE.

Materials and Methods

2.1 The group of selected teachers is very mixed: gender/age/subject? Can you report (and comment) this for the three groups as females often tend to be more critical with oneself? This is also relevant as you report results for men and female teachers separately (pg 6).

Please also check what you state concerning subject, this may be the whole sample?

2.2 Can you provide an argument for the date of the research, 2015?

2.3 For me it’s not clear who (and how) is drawing the thematic analysis, the whole group, one researcher? How did you organize the comparison of meaningful quotes? Did you have a small setup of meaningful terms, concepts, notions from the beginning as you stated “some units of recording (phrases/words), were highlighted in 141 colour, which seemed relevant to the study purposes” (pg 4, ln 141/142). It may be helpful to get a more detailed impression about that.

Please check reference style (pg 4, ln 146): 22 > Flick, 2005

Table 2 seems not be in line with the text. Please check dimensions/categories, perhaps giving clear characters/numbers to each of them?

Results

3.1 “specific equipment” (pg 7, ln 295): here it may be reasonable to refer to scholars criticizing PE profession for being narrow (Gray et al., 2015) and gendered (Camacho-Miñano et al., 2021). What are your thoughts on that?

Please check italic style for quotes.

Conclusions

“The results of this study support the importance of having good experiences in PE to generate positive attitudes towards PE” (pg 9, ln 414/415): from a more critical perspective, “sports games and competition for male teachers, and gymnastics and dance for women”. Could you elaborate more on this with regard to education. From my point of view, such experiences support a kind of gendered PE. What about education and/or inclusion?

Edits in terms of flow

Pg 1, ln 39: check spaces between and PE

Pg 2, ln 46: attach > do you mean combine with, link to…?

Pg 2, ln 52: … can exert influence ON their…?

Pg 3, ln 96/97: answers

Pg 3, ln 119: past their experiences > their past experiences

Pg 3, ln 121: scholar curricula > school curricula

Pg 4, ln 133: The research controlled > The research group/ a member of the research group controlled (maybe one researcher as you have a training before)?

Pg 4, ln 143: comple > check for…?

Pg 4, ln 144: check spaces records and of

Pg 4, ln 169: categories system > category system?... there might be even more, please check carefully!

References

Camacho-Miñano, M. J., Gray, S., Sandford, R., & MacIsaac, S. (2021). Young women, health and physical activity: tensions between the gendered fields of Physical Education and Instagram. Sport, Education and Society, 27(7), 803-815. doi:10.1080/13573322.2021.1932455

Gray, S., MacIsaac, S., & Jess, M. (2015). Teaching ‘health’ in physical education in a ‘healthy’ way. RETOS:

Nuevas tendenies en Educacion Fisica Deportes y Recreacion28, 165-172.

Author Response

(The authors gave the same response as above.)

Round 2

Reviewer 1 Report

Thanks you for all changes and explanations.